# Glucocorticoid Hormones as Modulators of the Kynurenine Pathway in Chronic Pain Conditions

**DOI:** 10.3390/cells12081178

**Published:** 2023-04-18

**Authors:** Filip Jovanovic, Visnja Jovanovic, Nebojsa Nick Knezevic

**Affiliations:** 1Department of Internal Medicine, Merit Health Wesley, Hattiesburg, MS 39402, USA; filip.jovanovic@merithealthwesley.com; 2Department of Anesthesiology, Advocate Illinois Masonic Medical Center, Chicago, IL 60657, USA; visnja.jovanovic.phd@gmail.com; 3Department of Anesthesiology, University of Illinois, Chicago, IL 60612, USA; 4Department of Surgery, University of Illinois, Chicago, IL 60612, USA

**Keywords:** chronic pain, kynurenine pathway, HPA axis, tryptophan metabolism, glucocorticoids, cortisol, tryptophan 2,3-dioxygenase (TDO), indoleamine 2,3-dioxygenase (IDO)

## Abstract

The pathogenesis of chronic pain entails a series of complex interactions among the nervous, immune, and endocrine systems. Defined as pain lasting or recurring for more than 3 months, chronic pain is becoming increasingly more prevalent among the US adult population. Pro-inflammatory cytokines from persistent low-grade inflammation not only contribute to the development of chronic pain conditions, but also regulate various aspects of the tryptophan metabolism, especially that of the kynurenine pathway (KP). An elevated level of pro-inflammatory cytokines exerts similar regulatory effects on the hypothalamic–pituitary–adrenal (HPA) axis, an intricate system of neuro–endocrine–immune pathways and a major mechanism of the stress response. As the HPA axis counters inflammation through the secretion of endogenous cortisol, we review the role of cortisol along with that of exogenous glucocorticoids in patients with chronic pain conditions. Considering that different metabolites produced along the KP exhibit neuroprotective, neurotoxic, and pronociceptive properties, we also summarize evidence rendering them as reliable biomarkers in this patient population. While more in vivo studies are needed, we conclude that the interaction between glucocorticoid hormones and the KP poses an attractive venue of diagnostic and therapeutic potential in patients with chronic pain.

## 1. Introduction

The International Association for the Study of Pain (IASP) defines pain as “an unpleasant sensory and emotional experience associated with, or resembling that associated with, actual or potential tissue damage” [1]. Pain is considered a subjective experience, perceived and influenced by biological, psychological, and social variables of an individual, though the discovery of biomarkers for pain has recently challenged this definition. While acute pain serves as an adaptive behavior that protects from further tissue damage, an emerging body of evidence points out that this model of behavior includes not only simple avoidance from harm but complex neural pathways that predict any potential pain, mobilizing metabolic energy as needed and adjusting behavior before critical tissue damage occurs [2]. Following an injury, the nociceptive system undergoes structural changes [3]. Once the firing threshold of nociceptors becomes augmented, these receptors react more promptly to both painful and non-painful stimuli. Taken together, these changes serve to preserve, prevent, and protect the integrity of the tissue. Still, when structural plasticity turns maladaptive, it may lead to the genesis of chronic pain. Rather than just a continuation of acute pain, chronification implicates structural plasticity and reorganization involving both the nervous and the immune systems [4,5]. The effects of these changes, however, go far beyond the two systems—reaching the endocrine front as well.

Approaching pain from a wider perspective of nervous–endocrine–immune ensemble may be vital for a better understanding of this complex phenomenon. It has been hypothesized that dysregulation in any segment of this overarching supersystem plays a role in the pathogenesis of chronic pain [6]. In this respect, we seek to analyze pain persistence through the prism of a major metabolic route—that of the tryptophan–kynurenine pathway (KP). Tryptophan (TRP) is an essential amino acid of invaluable metabolic importance, especially necessary for the synthesis of proteins and select neurotransmitters. Different metabolic routes of TRP—such as kynurenine (KYN), serotonin, and indole pathways—have been of interest to the scientific community for decades. Moreover, published data have rendered the KP pivotal in numerous endocrine pathologies and different chronic pain conditions [7].

Following the suggestion that nervous, immune, and endocrine systems function interdependently [6,8,9], we analyze chronic pain in light of interactions among the 3 overlapping systems. Since pro-inflammatory cytokines have been linked to changes in the hormonal milieu of patients experiencing chronic pain, we concentrate on the effects that pro-inflammatory cytokines exert on the hypothalamic–pituitary–adrenal (HPA) axis. The relevance of this axis lies in that it poses a major mechanism of the stress response: to counter inflammation, the HPA axis secretes a type of steroid hormones called glucocorticoids. Cortisol—produced by the adrenal gland—is the most abundant endogenous glucocorticoid in humans [10,11]. The purpose of this review is to explore the interactions and importance between glucocorticoid hormones, on the one hand, and metabolites and enzymes of the KP, on the other, in different chronic pain pathologies.

## 2. Pathomechanisms of Chronic Pain

Chronic pain—defined as pain persisting or recurring for longer than 3 months—affects an estimated 20.4% of the US adult population [4,12]. Masaru Tanaka et al. argue in favor of component-based pathomechanisms of chronic pain, where each pain mechanism—nociceptive, neuropathic, nociplastic, and psychogenic pain—corresponds to a certain component in the pain pathway [13]. The pain pathway begins with transduction, the process occurring when certain stimuli activate a specific type of sensory neurons termed nociceptors, to convert pain into neural signals [13,14]. The second component is induction (also referred to as conduction), in which the signals are relayed from the peripheral to the central neurons. The third component is transmission, the act of transferring the neural signals from the peripheral to the second-order neurons. The fourth component refers to modulation in which pain signaling is inhibited, while the descending inhibitory fibers are activated. The final component is perception, which takes place when third-order neurons communicate with the somatosensory cortex [13].

The development of chronic pain is not a purely neurobiological phenomenon: a biopsychosocial approach puts into perspective the role of physiological, psychological, and social factors in the genesis of chronic pain syndromes [15]. Among the plethora of critical aspects, highly relevant to this manuscript is the interplay between the nervous and the immune systems. The transition from acute to chronic pain involves not only the neural network but a series of mediators such as non-neuronal (microglia, astrocytes, and Schwann cells) and inflammatory (neutrophils, macrophages) cells along with various cytokines, vasoactive mediators, chemokines, and their respective receptors [16]. While detrimental to physical and emotional well-being, chronic pain conditions can also alter the defensive capacities of an individual [17]. This is mainly because chronic pain goes hand in hand with inflammation.

Injury and the accompanying tissue inflammation induce a series of chemical reactions that trigger secretion of a range of signaling molecules—such as neurotransmitters, peptides, eicosanoids, neurotrophins, chemokines, and cytokines—making what is collectively referred to as the “inflammatory soup” [18]. Worth elaborating on are cytokines, which are polypeptides acting as mediators that control growth, differentiation, and function of various classes of cells [19]. Although involved in multiple physiological processes, cytokines are most prominent as agents modulating immune and inflammatory responses [20]. Depending on the function, cytokines may be generally characterized as pro- or anti-inflammatory. The plasma levels of pro-inflammatory cytokines interleukin-1β (IL-1β), interleukin-2, interleukin-6, interferon-γ (IFN-γ), and tumor necrosis factor-α (TNF-α) in particular have been found to correlate with pain intensity in patients suffering from chronic pain conditions [21]. An elevated level of pro-inflammatory cytokines resulting from either stressful stimuli or inflammatory states has been shown to create an imbalance among the different pathways of the TRP metabolism [13,22].

## 3. The Metabolic Pathways of TRP: KP and Serotonin Pathway

In reviewing the patterns of interaction between the nervous and the immune system that occur in pain and inflammation, Pinho-Ribeiro et al. demonstrated that inflammation contributed to the pathogenesis of numerous diseases and served as a key element in those causing both acute and chronic pain [14]. Chronic low-grade inflammation has been shown to activate the TRP metabolism [23]. TRP is an essential amino acid that plays various important roles in the human body, including synthesis of bioactive substances [24]. TRP metabolism has several pathways that produce KYN, serotonin (5-hydroxytryptpamine; 5-HT), and indole derivatives [25]. The KP poses the main branch of TRP metabolism, responsible for the catabolism of approximately 95% of available TRP [23]. The KP is a major catabolic route of TRP that results in the formation of diverse bioactive metabolites. The initial and rate-limiting step of this pathway begins with the conversion of TRP into N-formyl-L-kynurenine via the enzymes indoleamine 2,3-dioxygenase 1 (IDO-1), 2 (IDO-2), and tryptophan 2,3-dioxygenase (TDO) [26]. N-formyl-L-kynurenine is subject to the enzymatic activity of kynurenine formamidase in the production of KYN, which represents the central metabolite in this complex pathway. From here on, KYN can further be catabolized in 3 different directions, generating anthranilic acid (ANA; hydrolysis via kynureninase) [27], kynurenic acid (KYNA) (irreversible transamination via kynurenine aminotransferase [KAT] isoenzymes) [28], and 3-hydroxykynurenine (3-HK) (hydroxylation via kynurenine 3-monooxygenase) [29]. The 3-HK metabolite catabolizes either into xanthurenic acid (XA) (via KAT enzyme) or 3-hydroxyantranilic acid (via kynureninase). The final step of this metabolic cascade involves the activity of 3-hydroxyanthranilate dioxygenase (3-HANA) that catalyzes the production of quinolinic acid (QA). The alteration of the KP is known to lead to a series of cancerous, immune, neurodegenerative, and psychiatric pathologies [30]. The KYN and serotonin metabolic pathways of tryptophan are presented in Figure 1.

On the endocrine front, TRP partakes in the synthesis of serotonin and melatonin. [24] TRP is converted into serotonin by the processes of hydroxylation (via tryptophan hydroxylase [TPH]) and decarboxylation (via monoamine decarboxylase (5-hydroxytryptophan decarboxylase)) [24,31]. TPH is present in the form of two isoforms, tryptophan hydroxylase 1 and 2, which can be found in the gastrointestinal tract and the central nervous system (CNS), respectively [31,32]. Serotonin, therefore, does not only have a twofold biosynthetic pathway, but performs a double function—as a neurotransmitter and a peripheral hormone [33]. What is more, serotonin serves as a precursor to melatonin, which is produced via the enzymes N-acetyltransferase and N-acetylserotonin O-methyltransferase [34]. Released primarily from the pineal gland, melatonin plays a role in the regulation of sleep, the circadian rhythm, and immune system, but also in the modulation of the HPA axis [24,35].

Of particular interest is the mechanism behind the interaction of the KYN and the serotonin pathway. The pro-inflammatory cytokines activate the enzymes IDO-1 and IDO-2 that are generally expressed in non-hepatic tissues [36]; the TRP metabolism is then relocated away from the liver cells, where TRP is physiologically degraded, to non-hepatic tissues [22]. In mammalian models, it was shown that inflammation can trigger an increased adrenal production of cortisol, which activates the TDO enzyme [37]. The activation of the IDO enzymes (through the aforementioned pro-inflammatory cytokines) and the TDO enzyme (through inflammation-induced cortisol secretion) shifts the metabolism of TRP to the KP away from the competing serotonin pathway. Consequently, a surge in the production of KYN lowers the availability of TRP for the synthesis of serotonin [22]. The alteration of the KP, therefore, negatively impacts the TRP–serotonin pathway, and contributes to the development of depression [38].

### KP in Chronic Pain and Comorbid Depression

The levels and ratios between different metabolites of the KP have been implicated in comorbid chronic pain and depression [39]. A large meta-analysis of 101 studies elaborated on the shift of TRP metabolism from serotonin to the KP and demonstrated decreased TRP and KYN across major depressive disorder (MDD), bipolar disorder, and schizophrenia [40]. The cellular mechanism behind anxio-depressive comorbidities in chronic pain disorders has been mainly attributed to the serotonergic pathway that operates among the dorsal raphe nucleus, the central amygdala, and the lateral habenula [41]. An MRI analysis of human brain confirmed the findings suggested by murine models of this circuit’s involvement in depression and pain: patients with chronic back pain and comorbid depression have a reduced connectivity between the dorsal raphe nucleus and the central amygdala—unlike healthy controls or patients with chronic back pain who do not experience depressive symptoms [42]. Substance abuse has also been evaluated in depression pathogenesis [43]. It was found that the use of exogenous cannabinoids, including tetrahydrocannabinol and its derivatives, was associated with upregulated pro-inflammatory cytokines and decreases TRP and serotonin. In demonstrating that depressive symptoms secondary to chronic pain cellularly differ from primary depression, the circuit between the three aforementioned brain structures opens up new possibilities for various therapeutic and pharmacological interventions targeting serotonergic activity.

## 4. Pain Quantification Using Biomarkers

The KP is intricately connected with chronic pain given its activation via pro-inflammatory mediators in patients experiencing chronic pain conditions [7,44]. The research community largely believed that pain perception could not be objectively measured [45], as it was long deemed a “subjective” awareness [46,47]. What additionally reinforced such belief was the prevalence of comorbidities, on the one hand, and the complex origins of chronic pain, which involve biological, psychological, and even social factors, on the other [44]. Recent scientific advances, however, refute the claim about the nonquantifiable nature of pain owing to the discovery of biomarkers for painful conditions [44]. Biomarkers are objectively measured parameters that can be utilized as precise clinical tools in assessing physiological and pathophysiological processes, but also the effectiveness of therapeutic interventions [48]. Evaluating the role of different bioactive metabolites synthesized along the KP could be crucial in estimating patients’ pain levels.

### 4.1. KYN Metabolites as Biomarkers for Chronic Pain

Gunn and colleagues retrospectively studied pain-specific biomarkers on a cohort of 17,834 individuals undergoing opioid treatment for chronic pain [44]. Of the 11 biomarker tests, as many as 3—QA, KYNA, and XA—were metabolites of the KP; meanwhile, serotonin was the only examined metabolite of the TRP–serotonin pathway. Atypical metabolite levels were most frequently observed with QA and KYNA, which were elevated in 29% and 27% of samples, respectively. The studied pain population also exhibited abnormally high values of XA, which was elevated in 17% of samples. Considering that the activity of the TRP–serotonin pathway is inversely proportional to that of the KP, it is not surprising that decreased concentrations of serotonin were noted in 7% of samples. A similar retrospective observational study—though of much smaller scope (n = 298)—was conducted by Pope et al. [49]. Relevant to this review is their investigation of the KP biomarkers in the population experiencing unspecified chronic pain symptoms. Urine samples of patients living with chronic pain revealed high concentrations of QA in 36%, KYNA in 33%, and XA in 33% of cases. Furthermore, a study by Groven et al. measured certain KYN metabolites and their ratios in female patients suffering from chronic fatigue syndrome (n = 49) and fibromyalgia (n = 57), and the results were in alignment with the aforementioned biomarkers of chronic pain. Not many papers have tackled how pro-inflammatory cytokines regulate TRP metabolism in the HPA axis. A rare example is an in vitro study by Tu and colleagues, who used murine models to address neuroendocrine aspects of cytokine-induced illnesses in the context of TRP degradation [50]. The authors found that IFN-γ induces IDO expression in the cells of hypothalamic (GT1-7) and pituitary origin (AtT-20), suggesting that this may contribute to accumulation of the KP’s toxic metabolites, particularly QA and 3-HK. However, this in vitro experiment demonstrated that IFN-γ did not activate IDO in adrenal cells (Y-1).

### 4.2. KYN Metabolites and the Blood–Brain Barrier

The physical interface between the peripheral circulation and the CNS exists to maintain the narrow homeostatic range of the neural environment. The elements of this interface include: (1) the blood–brain barrier (BBB) that is comprised of microvascular endothelial cells separating cerebral interstitial fluid and blood; (2) the blood–cerebrospinal fluid barrier containing the choroid plexus epithelial cells responsible for the production of cerebral spinal fluid (CSF); and (3) the avascular arachnoid epithelium that parts the blood and the subarachnoid CSF [51]. The integrity of the BBB is maintained by the *neurovascular unit*, a complex consisting of capillary endothelial cells, the capillary basement membrane, pericytes, astrocytes, neurons, and microglia. When discussing how certain KYN metabolites cross the BBB in chronic pain conditions, one should also factor in that inflammation alters the permeability of the BBB [52].

In 1991, Fukui et al. measured the rates of brain uptake of different metabolites of the KP [53]. The authors showed that a neutral amino acid carrier of the BBB facilitates the uptake of KYN and 3-HK, whereas QA, KYNA, 3-HANA, and in particular ANA cross the BBB by passive diffusion. These findings provided inference that peripheral KYN and possibly 3-HK contribute to their corresponding cerebral pools. A study with Swiss-Webster mice demonstrated that intraperitoneal application of KYN and probenecid resulted in significantly increased KYNA cerebral concentrations and produced anticonvulsant effects in a dose-dependent fashion [54]. The effects of probenecid, a known organic acid transport inhibitor from the CSF, on the increased cerebral content of KYNA (up to 1300-fold from baseline concentrations) have been previously described [55,56]. A follow-up study validated these results by showing that administration of KYN and probenecid exerted protective effects against pentylenetetrazol-induced seizures from both behavioral and electrophysiological aspects [57]. It was not long before a new compound, glucosamine-KYNA, was synthesized and compared to pure KYNA on the hippocampal-evoked activity in male Wistar rats [58]. The drugs were administered alone or in combination with probenecid, and the results were consistent with reduced glutamatergic input in the CA1 region of the hippocampus with the glucosamine-KYNA, but not with pure KYNA. This effect was augmented with the administration of probenecid, thereby reinforcing the findings from previous studies.

### 4.3. KYNA as a Target Compound in Nociceptive Processing

The knowledge behind the mechanism of different metabolites of the KP in the pathogenesis of chronic pain continues to evolve; still, compounds such as KYNA have been attracting considerable attention in this context. Location plays a role in the production of KP’s neuromodulating metabolites—the neurotoxic branch seems to be separated from the neuroprotective one. On the one hand, an in vitro study on human fetal microglia and adult macrophages, activated in the pathophysiology of multiple neurological conditions, showed that both of these immune cells are instrumental in the synthesis of neurotoxic QA [59]. On the other hand, a similar study revealed that astrocytes, specialized glial cells, serve as producers of generally neuroprotective KYNA [60]. KYNA is a nonselective antagonist of three ionotropic receptors: the kainic acid receptor, the strychnine-insensitive glycine-binding site of the N-methyl-D-aspartate (NMDA) receptor, and the α-amino-2,3-dihydro-5- methyl-3-oxo-isoxazolpropionic acid (AMPA) receptor [61]. Interestingly, KYNA has been found to exhibit a concentration-dependent neuromodulatory effect by facilitating (nanomolar concentration) or inhibiting (micromolar concentration) the NMDA and AMPA receptors [62,63].

From a pain perspective, not only may KYNA have an important role in ameliorating nociception through NMDA receptor modulation, but it also influences peripheral (sensory nerve fibers) and other central (dorsal horn and glial cells) components of the nervous system. For example, a study showed that intrathecal injection (45 μg) of KYNA resulted in a dose-dependent depression of the strychnine-evoked hyperesthesia through NMDA antagonism in rat models [64]. Chapman and Dickenson showed that coadministration of 2.5 μg 7-chlorokynurenate (7 CK) and 5μg morphine resulted in a 91% and 94% decrease of the neuronal input and frequency-dependent potentiation of dorsal horn nociceptive neurons (wind-up), respectively [65]. The authors proposed that NMDA-mediated wind-up may play a role in central pain processing. An in vivo experiment by Schneider and Perl explored the effects of KYNA on cutaneous mechanical stimulation in male hamsters [66]. The A- and C-fiber components of the afferent nerves activated by L-glutamate were distributed within the superficial dorsal horn (laminae I and II). The responses from certain dorsal horn neurons receiving inputs from these fibers were antagonized by KYNA (1 μg) administration. A study with an inflammatory model of pain investigated the relationship between 7 CK and dorsal horn neuronal responses after peripheral 5% formaldehyde administration [67]. The authors showed that intrathecal administration of 7CK (0.25–2.5 μg) had no effect on the first phase of the formalin response; however, the reverse was true for the second phase, implicating that the NMDA receptor induced and maintained the second phase in the inflammatory pain model. Christoph et al. further validated the antagonistic properties of 5,7-dichlorokynurenic acid on peripheral NMDA receptors in the formalin test and two models of neuropathic pain, thereby eliminating serious centrally mediated side effects seen in different NMDA antagonist classes [68]. Ma and Zhao performed a tetanic stimulation of the sciatic nerve in rats and induced long-term potentiation of C-fiber potentials in the spinal dorsal horn [69]. The application of a glial metabolic inhibitor, intrathecal fluorocitrate (1 nmol), resulted in long-term depression; however, these findings were abolished by intrathecal KYNA (50 nmol), suggesting that glial cell modulated central sensitization of nociceptive neurons through the actions of NMDA receptors. Edwards et al. investigated the effects of diclofenac or exposure to noxious stimuli (tail ischemia) on concentrations of glutamate and KYNA in rat models in CNS regions [70]. Both noxious stimulation and nonsteroidal anti-inflammatory drugs (NSAIDs) increased glutamate concentrations in the diencephalon and spinal cord. Similarly, the KYNA concentrations were particularly elevated in the midbrain (ischemia group) and lumbo-sacral spinal cord regions (diclofenac group), implicating that KYNA synthesis is a prostaglandin-independent antinociception mechanism of NSAIDs. Csáti et al. highlighted the importance of glutamate receptors by using complete Freund’s adjuvant (CFA) to elicit activation of trigeminal ganglion, a finding reversible with KYNA administration [71].

### 4.4. The KP in Migraine and Neuropathic Pain

Migraine and neuropathic pain, despite clinically diverse manifestations, are neurological conditions that share a common etiology. A study on human subjects confirmed a molecular imbalance in the KP and serotonin pathway among female patients with migraine, identifying persistently suboptimal concentrations of select TRP metabolites as a possible attack trigger [72]. Conversely, neuropathic pain stems from oversensitization of the NMDA glutamate receptors [73]. Neuromodulating compounds produced along the KP partake in the pathogenesis of migraine through their interference with the trigemino-vascular system and glutamatergic receptors in the CNS [73,74]. A study on orofacial pain examined the concentrations of glutamate, KYNA, and its precursor KYN in male Sprague Dawley rats injected with CFA: the findings established glutamate as a neurotransmitter highly influential in early pain processing and confirmed that the compensatory synthesis of KYNA acts antagonistically towards the glutamatergic receptors [75].

Most research endeavors in the domain of peripheral and central pain processing focus on the modifying effects that KYNA has on glutamate and its receptors [76]. Knyihár-Csillik and colleagues analyzed the analgesic effects of a KYN/probenecid combination in an experimental migraine model [77]. Through electrical stimulation of the trigeminal ganglion, the authors showed an immunohistochemical localization of c-fos in second-order sensory cells in the caudal trigeminal nucleus. However, pretreatment with KYN/probenecid prevented such increase of c-fos immunoreactive cells and prevented the propagation of noxious stimuli from the brain stem to the thalamus. Heyliger and colleagues likewise investigated possible analgesic properties of TRP and its metabolites by injecting male Sprague Dawley rats with different metabolites of the KP [78]. Pain sensitivity was measured before and after the compound administration using the hotplate and tail-flick tests. TRP, KYN, KYNA, and QA were shown to induce analgesia, though its strength and persistence differed depending on the compound, the dosing, and the method used. In the context of migraine aura and KCl-induced cortical spreading depression (CSD), among male and female rats, systemic pretreatment with KYN, probenecid, and KYN–probenecid resulted in an increased cortical KYNA level [79]. The ability to attenuate the CSD was observed with all 3 treatment arms in female rats (especially in the diestrus stage of the estrus cycle), as compared with only KYN–probenecid in male rats. What is more, research has shown that CSD increases the permeability of BBB, and systematically administered KYNA ameliorates the effects of CSD [80].

### 4.5. The KP Compounds as Analgesic Targets in Chronic Pain

Ciapała et al. discussed how our theoretical and practical knowledge of the KP translates into therapeutic potential, especially in the context of neuropathic pain. On the whole, 3 major mechanisms stand out as attractive sights for treating chronic pain: (1) KMO inhibition; (2) IDO inhibition; and (3) administration of KYNA or KYNA-related substances [52]. An extensive review evaluated inhibitors of the KP as neurotherapeutics [81]. Even though there was ample information on studies investigating different treatment options in cancer patients, data regarding pain were still limited. KYN metabolites have also been proposed as novel therapeutic targets in tackling migraine and neuropathic pain [82]. A review by Boros and Vécsei focused on preclinical and clinical use of 4-Cl-KYN, an L-KYN derivative, in neuropathic pain, depression, and suicide prevention [83]. 4-Cl-KYN is rapidly absorbed through the gut and readily penetrates the BBB, after which it is converted into 7-Cl-KYNA. This metabolite is a selective NMDA receptor antagonist and has 20 times affinity that of KYNA [84]. AV-101 (4-Cl-KYN) treatment with 360, 1080, and 1440 mg daily for 2 weeks has been shown to modulate allodynia, heat, and mechanical hyperalgesia secondary from capsaicin injection, although the results were not statistically significant [85]. A more recent study with constriction injury of the sciatic nerve in mice provided evidence that intrathecal administration of 1-D-MT (IDO inhibitor), UPF468 (KMO inhibitor), or L-KYN decreased mechanical and thermal hypersensitivity [86]. Targeting enzymes and metabolites of the KP is an avenue pharmacologically worth pursuing owing to our theoretical knowledge on the matter and numerous preclinical studies with promising results. Nevertheless, the end-goal would involve the development of drugs with higher selectivity, increased potency, a favorable side-effect profile, altogether with an emphasis on the genetic background of an individual to provide personalized treatment options.

## 5. The Effects of Chronic Pain on the HPA Axis

The HPA axis refers to an intricate system of neuro–endocrine–immune pathways and feedback loops that operate not only between the hypothalamus, pituitary gland, and adrenal glands but also with different nervous and immune cells [6,87]. As noted, the HPA axis constitutes a major mechanism of the stress response [10], and exposure to a stressor such as inflammation activates it as a part of adaptation initiated by the pain system [88]. The most important anatomic structures implicated in the stress response include the hypothalamic paraventricular nucleus, the anterior lobe of the pituitary gland, and the adrenal gland [89]. These structures collectively make the HPA axis, the regulation of which is subject to inhibition from circulating glucocorticoids. The PVN is composed of 3 neuronal types, including parvocellular, magnocellular, and long-projecting neurons.

The initial hierarchical level in the HPA axis commences with the release of corticotropin-releasing hormone (CRH) from the terminals of parvocellular neurons into the anterior pituitary lobe, adenohypophysis, which is divided into pars distalis, pars intermedia, and pars tuberalis. Of particular interest is the pars distalis, which produces thyroid-stimulating hormone, prolactin, growth hormone, follicle-stimulating hormone, luteinizing hormone, and adrenocorticotropic hormone (ACTH). Under the influence of different pro-inflammatory cytokines, this multistep process leads to the production of glucocorticoids [11]. The most abundant glucocorticoid hormone in humans is cortisol, whereas in rodents it is corticosterone. While generally part of an adaptive process, the prolonged activation of the HPA axis may turn maladaptive; in other words, inadequate control of the stress response can lead to the onset of various illnesses [90]. From a diagnostic standpoint, measuring the effects of the dysregulated HPA axis on a molecular and cellular level can be used to determine the presence and intensity of chronic pain [88].

### 5.1. Diurnal Cortisol Slope

Cortisol is a potent marker for psychoneuroendocrinological research, which can be quantified using blood, urine, saliva, or hair samples. In healthy individuals, cortisol levels rise around 50% to 60% upon waking up and reach a peak after 30 to 40 min [91]. Following the morning surge, cortisol levels first decrease rapidly and then more gradually throughout the day, hitting an all-day low at bedtime [92]. This pattern is referred to as diurnal cortisol slope. Measured under basal conditions, glucocorticoid hormones are secreted in accordance with the circadian and ultradian rhythms [93], both of which are regulated by the mineralocorticoid receptor (MR) [90]. Another noteworthy receptor responsible for glucocorticoid signaling in the brain and pituitary gland is the glucocorticoid receptor (GR) [90]. Although the GR has a lower glucocorticoid-binding affinity in comparison to the MR, the former is activated in the increased presence of glucocorticoids [94]. For this reason, the GR is believed to be accountable for the long-established inhibitory properties of glucocorticoids on inflammation [94,95]. Nevertheless, a prolonged exposure to pro-inflammatory cytokines can disrupt this signaling pathway by inhibiting the GR and altering its expression [96,97], leading to impaired negative feedback regulation of the HPA axis [98]. The major manifestation of this impairment is glucocorticoid resistance, which refers to decreased sensitivity to glucocorticoids [97].

Pro-inflammatory cytokines TNF-α, IL-1β, and interferon-α have been found to disrupt not only the HPA axis but to attenuate the mRNA expression of the so-called clock-controlled genes involved in the regulation of the diurnal cortisol slope [92,99,100]. Flattening of the diurnal wake-to-bedtime slope, which refers to a smaller decrease between the morning and evening cortisol levels, has been associated with poorer outcomes in immune and inflammatory diseases—as presented in a large systematic review and meta-analysis by Adam et al. [92]. Furthermore, an attenuated diurnal cortisol slope was also observed in patients whose chronic pain originates from chemotherapy-induced peripheral neuropathy compared to healthy individuals [101]. A flattening trend of the diurnal cortisol levels has not only been observed in multiple inflammatory-related chronic conditions, but has been linked to a poorer illness prognosis [102].

### 5.2. Cortisol and Chronic Pain

Multiple studies have attempted to answer how cortisol levels correlate with chronic pain. Over the years, the results have been far from uniform. In this respect, it should be emphasized that the interplay between chronic stress, chronic pain, and serum cortisol is contingent on a number of factors. How stressful stimuli will modulate the HPA axis depends not only on the nature of the stressor but also on individual human predispositions [103,104]. The factors impacting cortisol levels in patients with chronic pain include, but are not limited to, pain intensity, pain duration, the use of medications (especially opioids and antidepressants), concurrent depression and/or anxiety, smoking status, body mass index (BMI), physical inactivity, and sleep deprivation [104]. Given that even childhood exposure to pain has a lifelong impact on a person’s pain programing and their neuroendocrine system [88], it is exceedingly difficult to account for all the factors impacting cortisol levels in patients with chronic pain.

These theoretical statements were practically assessed by Van Uum et al., who investigated the role of cortisol as a biomarker for long-term stress [104]. Cortisol levels were elevated in the hair samples of patients with non-cancer chronic pain (n = 15) who were treated with opioids for at least a year, but who were not on any hormone therapy at the time of sample collection, in comparison to healthy controls (n = 39). The included patients with chronic pain suffered from lumbar degenerative disc disease, chronic muscular pain, chronic neuropathic pain, or chronic visceral pain [104]. Another study, conducted by Begum et al., measured morning and evening cortisol levels from the saliva of at-risk patients with acute pain (n = 20) and of those patients with chronic pain (n = 19) who were not taking any centrally acting medications known to interfere with the HPA axis (e.g., opioid analgesics)—therefore removing the limitations of previous research where subjects with chronic pain were on opioid therapies [105]. Much higher levels of both morning and evening cortisol were detected in the groups with acute and chronic pain relative to controls without pain.

### 5.3. Glucocorticoids and the KP

More than 40 years ago, a study was conducted by Danesch and colleagues to evaluate how a synthetic glucocorticoid influences the expression of TDO mRNA [106]. The authors injected the adrenalectomized rats with dexamethasone (10 μg/100 g of body weight) vs. normal saline in controls. Following an injection, there was a 10-fold increase in TDO, although no change in the amount of albumin mRNA was observed. It was concluded that dexamethasone exerted the effects on the rate of transcription, but not translation, of the TDO mRNA. In addition, a follow-up study by Nakamura et al. elaborated on the existence of a short-lived mediator protein, which was deemed important in evaluating the stimulatory effects of dexamethasone on the transcription activity of the TDO enzyme [107]. A pioneer work by Hirota et al. discovered a novel glucocorticoid-binding protein involved in the TDO enzyme induction [108,109]. In their experiment, rats were injected with different doses of dexamethasone, and liver cytosol was obtained 20 h following decapitation [108]. Not only did the authors provide direct evidence of a new dexamethasone-binding protein, labeled *peak C*, but they also showed that the dose of dexamethasone needed to be significantly higher to induce TDO (≥20 μg/100 g body weight) compared to tyrosine aminotransferase (≥2 μg/100 g body weight). The new highly differentiated GR (peak C) was associated with maximal induction of the TDO enzyme.

The same enzyme was of particular interest to Green and colleagues, who investigated the effects of TRP concentration, hydrocortisone, and allopurinol administration on TDO enzymatic activity [110]. Relevant to the scope of the present review, the cited authors showed a fourfold increase in the TDO accompanied by upregulated KYN production. What is more, Altman and Greengard developed an assay for measuring the TDO activity in human liver biopsy samples [111]. The authors had the study subjects injected with 250 mg of intramuscular hydrocortisone, which resulted in a twofold to fourfold increase in TDO activity, essentially rendering this enzyme inducible by hydrocortisone in vivo. These results were challenged by a more recent study that investigated TRP metabolism in the context of the HPA axis in a longitudinal cohort of patients with single MDD (n = 625) and recurrent MDD (n = 695) [112]. Elevated evening cortisol levels, consistently observed in patients with chronic depression, were shown to be associated with a decreased KYN/TRP ratio—a finding hypothesized to stem from downregulated GR activity secondary to high endogenous basal cortisol levels.

Another study was conducted with the intent of examining the pharmacokinetic and pharmacodynamic properties of cortisol and its relationship with the KP in patients with or without adrenal crisis receiving a lower (0.2–0.3 mg/kg/day) and higher (0.4–0.6 mg/kg/day) hydrocortisone substitution dose [113]. The patient group with adrenal crisis had both lower urinary cortisol and cortisone excretion, alongside significantly higher serum KYN levels (2.64 μmol vs. 2.23 μmol [*p* = 0.03]), and these findings were independent from the administered hydrocortisone dose. The authors rendered the efficacy of hydrocortisone reduced in patients with adrenal crisis, suggesting the need for a higher substitution dose to bridge the apparent lower cortisol sensitivity.

## 6. Discussion

Chronic pain has been linked to imbalances involving, not only neurological and immune, but also the endocrine system. In crude terms, pro-inflammatory cytokines synthesized in chronic pain conditions activate the HPA axis to secrete cortisol, which then modulates the expression of select enzymes of the KP. In order to facilitate our understanding of how different chronic pain conditions exert changes on the KP at the level of the HPA axis, it is important to know that depression and other mental health disorders often present in this patient population and reliably reflect the changes in the KP. This is particularly true when taking into account that inflammatory stimuli and stress hormones result in upregulation of the rate-limiting enzymes of the KP and induce depression-like behaviors. To understand how pro-inflammatory cytokines modulate the HPA axis, we sought to explain why the studies measuring cortisol levels have been reporting conflicting results.

First of all, we need to acknowledge the wide range of individual factors affecting cortisol levels in painful conditions. The most prominent ones include pain intensity, pain duration, pharmaceuticals, concurrent mental health conditions, smoking status, BMI, physical activity, sleeping pattern, and even childhood exposure to pain [88,104]. In the previously cited study by Van Uum et al., several factors could have contributed to the increased cortisol levels in patients with chronic pain as opposed to healthy controls. To begin with, as many as 67% of patients were smokers in comparison to 3% of controls; patients with chronic pain on average had a higher BMI than healthy individuals in the study; finally, all patients were taking opioids and more than half were taking an antidepressant [104]. It is not surprising that half of the subjects with chronic pain were taking an antidepressant given that anxiety and depression often accompany chronic non-cancer pain conditions [114], as explained through the interaction between the TRP–KYN and TRP–serotonin pathways. In short, when the KP is activated, the lack of TRP for the synthesis of serotonin occurs in the competing TRP–serotonin pathway. In line with these results is a recent experimental study that measured salivary cortisol in patients with chronic pain, which found that anxiety and depression scores corresponded to higher cortisol levels [105]. On the whole, the strength of Van Uum et al.’s study lies in that measuring cortisol from hair offers an opportunity to assess systemic levels of cortisol in the weeks and even months prior to sample collection; what is more, unlike saliva, cortisol found in hair is not subject to diurnal variations [104].

While it is important to account for individual factors influencing cortisol secretion in chronic pain diseases, it is also pertinent to seek a pathophysiological rationale behind this phenomenon. Begum and colleagues, who investigated salivary cortisol levels of patients with chronic pain and of those at risk, argued that elevated readings of morning and evening cortisol in the cohorts with acute and chronic pain stem from a pronounced physiological stress response [105]. This study also addressed the previously reported conflicting results of cortisol levels in patients experiencing chronic stress [115,116,117,118,119,120]. It retested the hypothesis formulated by Miller et al. that salivary cortisol is elevated at stressor onset but that it decreases as the time passes, eventually leading to hypocortisolism (Addison’s disease) [103,105]. Begum et al. found a high cortisol profile both in the group experiencing acute pain and in that experiencing chronic pain, thereby refuting Miller et al.’s hypothesis [105].

Cortisol was consistently increased in patients experiencing chronic pain in both Van Uum et al.’s and Begum et al.’s studies [104,105], though it is only the latter that gives us insight into the diurnal rhythm. High evening values obtained in the study measuring salivary cortisol and high overall values in both studies indicate flattening of the diurnal cortisol slope [104,105]. Such a blunted cortisol curve can be explained by glucocorticoid resistance, which occurs as a result of prolonged exposure to pro-inflammatory cytokines. Raison et al.’s research associates the flattening of the diurnal cortisol slope with poorer prognosis in patients chronically exposed to inflammation [102].

One of the first in vivo pieces of evidence illustrating the relationship between glucocorticoids and the rate-limiting TDO enzyme of the KP was demonstrated after adrenalectomized rats were injected with dexamethasone (10 μg/100 g of body weight) [106]. Following an initial lag phase of 30 min, the increasing mRNA TDO levels reached a 10-fold higher level four hours after the injection. A follow-up study using ≥20 μg/100 g body weight led to the discovery of a high-affinity dexamethasone-binding receptor, peak C, indicating that a much higher dose of dexamethasone was needed to induce maximal TDO activity [108]. It may appear at first that simply a higher corticosteroid dose is necessary to upregulate the TDO enzyme, although newer studies continue to explore the metabolic complexity behind this process. One such study by Brooks and colleagues investigated the interactions between pro-inflammatory mediators including IFN-γ, lipopolysaccharide (LPS), and polyinosine-polycytidylic and different corticosteroids (dexamethasone [12.5 μM], aldosterone [100 nM], and corticosterone [1 μM]) [121]. The authors developed an assay to evaluate the tissue expression and enzymatic activity of full-length (FL) and variant (v) transcripts of the enzymes IDO-1, IDO-2, and TDO. Validated organotypic hippocampal slice cultures that maintain corticosteroid responses occurring in intact rodent brain were used in the analysis. It was shown that the addition of the aforementioned pro-inflammatory cytokines did not alter the expression of the TDO transcript; still, dexamethasone and corticosterone induced the GR-mediated expression of both the TDO-FL and TDO-v transcripts. It was not surprising that the expression of TDO-FL/v transcripts was highest in the hippocampus, which features the highest density of GR and MR. In contrast, conditions with elevated basal cortisol levels such as depression have been associated with diminished activity of GR on TDO-expressing cell types, subsequently resulting in low TDO enzyme activity [112].

These conflicting results may be partially explained by polymorphisms in both the GR genes and the TDO promoter regions. Indeed, the sensitivity to glucocorticoids does exist among individuals, but is rather stable within an individual, and genetic predisposition for the setpoint of the HPA axis has been suggested [122]. Van Rossum and Lamberts discussed the existence of four polymorphisms of the GR gene: BclI and N363S (associated with hypersensitivity to glucocorticoids), ER22/23EK (associated with relative resistance to glucocorticoids), and TthIIII (no glucocorticoid sensitivity difference relative to other polymorphisms) [123]. Furthermore, in an in vitro study with A549 and HepG2 cells, a total of 12 polymorphisms, three of which in putative glucocorticoid responsive elements (GREs), were identified in the promoter region of the human TDO2 gene [124]. GREs are specific DNA sequences positioned in regulatory regions of target genes, important for potentiation or repression of transcription [125,126]. Promoter sequences are DNA segments located upstream of a gene where RNA polymerase and transcription factors initiate the transcription of the particular gene. The authors used four allelic variants (M1–M4) comprising 10 of the 12 identified polymorphisms, and promoter activities were evaluated under basal conditions, GR over-expression, or a GR over-expression and dexamethasone 1 μM exposure combination [124]. The results were consistent with increased promotor activities in HepG2 M1–M3 (GR over-expression) and A549 M1–M4 (GR over-expression and dexamethasone combination). The authors did not observe any correlation between the genetic variations and the TDO enzyme activity, albeit they suggested that a larger sample size would possibly challenge these findings.

While there is supporting evidence throughout literature that glucocorticoids upregulate the activity of the TDO enzyme, the data for the other rate-limiting enzymes of the KP—IDO-1 and IDO-2—continue to evolve. As mentioned, Tu et al.’s experiment involving the cells of the HPA axis confirmed that pro-inflammatory cytokine IFN-γ can upregulate IDO in hypothalamic and pituitary cells, thereby leading to the dysregulation of the whole HPA axis. Interestingly, the previously cited study also found that dexamethasone, corticosterone, and aldosterone can potentiate the ability of IFN-γ to induce the expression of IDO-1 FL/v, whereas only corticosterone was implicated in such process for the IDO-2/v enzyme [121]. These results imply that both GR/MR versus MR only are involved in the regulation of IDO-1 and IDO-2 enzymes of the KP, respectively; however, such results were unexpected in the context of anti-inflammatory properties of corticosteroids. The schematic interaction between different chronic pain conditions, the HPA axis, and the KP is presented in Figure 2.

The metabolites produced along the KP—such as QA, KYNA, XA, and 3-HK—have been shown to exhibit various properties, including those of neuroprotective, neurotoxic, and pronociceptive nature [127,128]. As shown in the study by Groven et al., the neuroprotective ratios of KYNA/QA and KYNA/3-HK were both reduced in patients with chronic fatigue syndrome and fibromyalgia, respectively, rendering these KYN metabolites potential biomarkers in the diagnosis and treatment of such chronic pain conditions [129]. The quantifiable findings of Gunn et al.’s, Pope et al.’s, and Groven et al.’s studies on pain-specific biomarkers involving metabolites of the KP are in line with our theoretical knowledge about the activation of the KP in response to the upregulation of pro-inflammatory cytokines provoked by chronic inflammation [44,49,129]. Overall, these 3 studies reinforce the stance that the metabolites formed down the KP—together with those modifiable by its activation—may be biomarkers of reliable sensitivity for chronic pain conditions. Empirical evidence demonstrating that accumulated byproducts of the KP can accurately quantify pain is a colossal step in the research of this pathway, which is expected to improve the monitoring and management of different chronic pain conditions.

## 7. Conclusions

This manuscript presents a link between the KP and cortisol as a chief endogenous hormone of the HPA axis in patients with chronic pain. In addition to reviewing in vitro studies assessing the effects of exogenous glucocorticoids, we also summarized the evidence suggesting that the metabolites of the KP can serve as reliable biomarkers for chronic pain. Through the analysis of the neuro–endocrine–immune system, we highlight the role of cortisol and exogenous glucocorticoids as promising modulators of the KP. While more in vivo studies are needed to supplement and fortify the findings of this review, we conclude that the interaction between glucocorticoid hormones and KP poses an attractive venue for diagnostic and therapeutic potential in patients with chronic pain.

## Figures and Tables

**Figure 1 cells-12-01178-f001:**
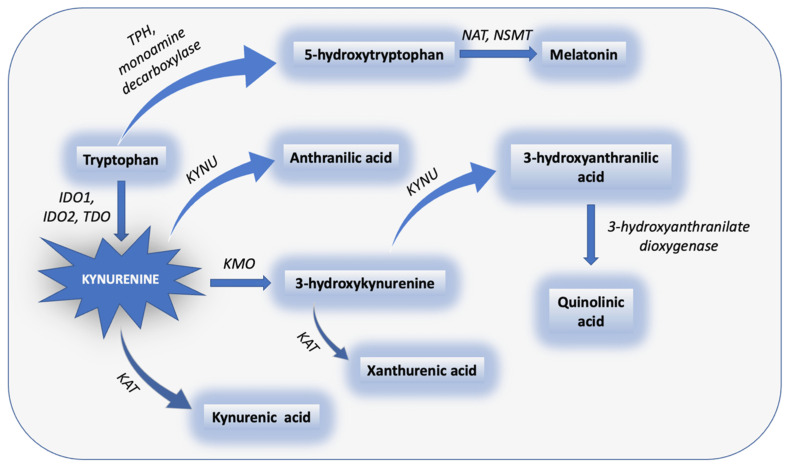
The kynurenine and serotonin pathway. Abbreviations: IDO1, indoleamine 2,3-dioxygenase 1; IDO2, indoleamine 2,3-dioxygenase 2; TDO, tryptophan 2,3-dioxygenase; KAT, kynurenine aminotransferase; KMO, kynurenine 3-monooxygenase; KYNU, kynureninase; TPH, tryptophan hydroxylase; NAT, N-acetyltransferase; NSMT, N-acetylserotonin O-methyltransferase.

**Figure 2 cells-12-01178-f002:**
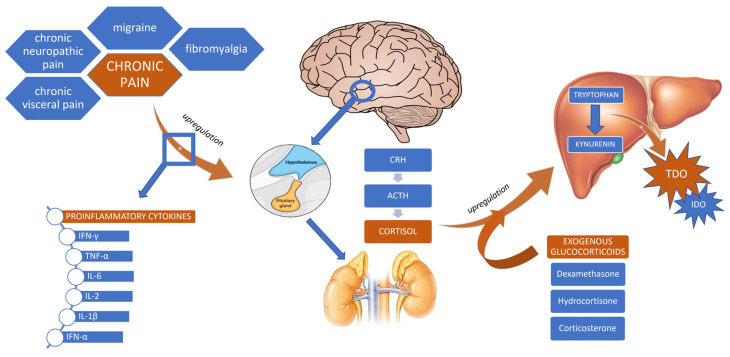
Cortisol and exogenous glucocorticoids as modulators of the KP in different chronic pain conditions. Abbreviations: IFN-γ, interferon-γ; TNF-α, tumor necrosis factor-α; IL-6, interleukin-6; IL-2, interleukin-2; IL-1β, interleukin-1β; IFN-α, interferon-α; CRH, corticotropin-releasing hormone; ACTH, adrenocorticotropic hormone; TDO, tryptophan 2,3-dioxygenase; IDO, indoleamine 2,3-dioxygenase.

## Data Availability

Not applicable.

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
