# Peer review of "Glucocorticoid Hormones as Modulators of the Kynurenine Pathway in Chronic Pain Conditions"

_cells, 2023, doi:10.3390/cells12081178_

Round 1

Reviewer 1 Report

The manuscript integrates information related to both the KP, the HPA axis, and glucocorticoid receptors with chronic pain. However, it is important to include a section with evidence from studies of patients with chronic pain disorders in relation to markers of inflammation and cortisol, as shown in Figure 2, to complement and strengthen the findings of this review.

Most of the evidence points to neuroinflammation and the KP pathway. A pathway between pain and depression was recently proposed. These articles could help to strengthen the relationship of chronic inflammation, and the involvement of tryptophan-kynurenine pathway metabolites.

Masaru Tanaka, Nóra Török, Fanni Tóth, Ágnes Szabó and László Vécsei,  Co-Players in Chronic Pain: Neuroinflammation and the Tryptophan-Kynurenine Metabolic Pathway

Biomedicines. 2021 Jul 26;9(8):897. doi: 10.3390/biomedicines9080897

Anke Tappe-Theodor, Rohini Kuner. A common ground for pain and depression. Nat Neurosci, 2019 Oct;22(10):1612-1614.  doi: 10.1038/s41593-019-0499-8.

Author Response

Reviewer’s Comment: “The manuscript integrates information related to both the KP, the HPA axis, and glucocorticoid receptors with chronic pain. However, it is important to include a section with evidence from studies of patients with chronic pain disorders in relation to markers of inflammation and cortisol, as shown in Figure 2, to complement and strengthen the findings of this review. Most of the evidence points to neuroinflammation and the KP pathway. A pathway between pain and depression was recently proposed. These articles could help to strengthen the relationship of chronic inflammation, and the involvement of tryptophan-kynurenine pathway metabolites.

Masaru Tanaka, Nóra Török, Fanni Tóth, Ágnes Szabó and László Vécsei, Co-Players in Chronic Pain: Neuroinflammation and the Tryptophan-Kynurenine Metabolic Pathway

Biomedicines. 2021 Jul 26;9(8):897. doi: 10.3390/biomedicines9080897

Anke Tappe-Theodor, Rohini Kuner. A common ground for pain and depression. Nat Neurosci, 2019 Oct;22(10):1612-1614.  doi: 10.1038/s41593-019-0499-8.”

Authors’ Responses: The authors wish to thank the Reviewer for their constructive feedback on the manuscript. The proposed changes have been made with the view of enriching the present review and fortifying its scientific value. The interventions encompass the following edits:

  • Section 1 KP in Chronic Pain and Comorbid Depression has been added to reference and explain the recently proposed pathway between chronic pain conditions and depression, which was previously not elaborated on in the manuscript. The suggested manuscript (Anke Tappe-Theodor, Rohini Kuner. A common ground for pain and depression. Nat Neurosci, 2019 Oct;22(10):1612-1614. doi: 10.1038/s41593-019-0499-8.) was included in this section.
  • The suggested paper (Masaru Tanaka, Nóra Török, Fanni Tóth, Ágnes Szabó and László Vécsei, Co-Players in Chronic Pain: Neuroinflammation and the Tryptophan-Kynurenine Metabolic Pathway. Biomedicines. 2021 Jul 26;9(8):897. doi: 10.3390/biomedicines9080897) was included in the manuscript with the aim of elucidating the complex relationship between inflammation and the KP metabolites in chronic pain conditions.
  • We agree with the Reviewer that it is important to provide information between inflammation, cortisol, and chronic pain disorders. We have included the studies by Van Uum et al., Begum et al., and Adam et al. to provide the evidence of patients with chronic pain disorders in relation to cortisol as a biomarker. We have also included the studies by Gunn et al., Pope et al., and Groven et al. have been referenced to provide the evidence of patients with chronic pain disorders in relation to inflammation as a biomarker.

Reviewer 2 Report

The topic of the review is significant and raises an interesting area of research, but the problem requires much deeper analysis and provding more specific details, in particular:

  1. 1. There is no description of the blood-brain barrier (BBB) and no information which KP metabolites can cross BBB – it must be mentioned and discussed while talking about chronic pain. 

  1. 2. There is no information about the exact place of action of the KP metabolites in the mechanism of pain (does it affect brain structures/spinal cord or periphery?) What is the exact effect on the structures of nervous system? - a figure/scheme could be added. 

  1. 3. There is no description of analgesic properties of the KP metabolites – a new chapter is required. 

  1. 4. The authors did not mention the KP metabolites and migraine-related mechanisms and neuropathic pain – a new chapter is required. 

  1. 5. Do the authors have a proposal for the use of drugs acting in the KP for chronic pain? 

  1. 6. There is no adequate references to related and previous work, e.g. the authors should mention scientific work of the Vécsei L. et al., Körtési T et al.

Author Response

  1. Reviewer’s Comment: “The topic of the review is significant and raises an interesting area of research, but the problem requires much deeper analysis and provding more specific details, in particular:

There is no description of the blood-brain barrier (BBB) and no information which KP metabolites can cross BBB – it must be mentioned and discussed while talking about chronic pain.”

  1. Authors’ Responses: The authors wish to thank the Reviewer for providing elaborate feedback on the manuscript and for giving constructive comments on the areas of this review that were in need of further refinement. We believe that these changes will help strengthen the present review and fortify its scientific value. The manuscript has been revised in accordance with the reviewers’ commentary and the suggested changes have been made. Please note that we have included the section on blood-brain barrier with a particular emphasis on which metabolites of the KP can cross BBB. The section is entitled 2. KYN Metabolites and the Blood-Brain Barrier.

  1. Reviewer’s Comment: “There is no information about the exact place of action of the KP metabolites in the mechanism of pain (does it affect brain structures/spinal cord or periphery?) What is the exact effect on the structures of nervous system? - a figure/scheme could be added.”

  1. Authors’ Responses: We agree that in order to better understand the role of KP in chronic pain conditions, there should be further discussion regarding the structures that KP metabolites act on. The section 3. KYNA as a Target Compound in Nociceptive Processing has been added with the intent of reviewing KYNA as chief KP metabolite involved in the mechanism of chronic pain.

  1. Reviewer’s Comment: “There is no description of analgesic properties of the KP metabolites – a new chapter is required.”

  1. Authors’ Responses: We thank the Reviewer for this valuable comment. A new section 5. The KP Compounds as Analgesic Targets in Chronic Pain was introduced and pertinent studies that have investigated analgesic properties of the KP metabolites in chronic pain were reviewed.

  1. Reviewer’s Comment: “The authors did not mention the KP metabolites and migraine-related mechanisms and neuropathic pain – a new chapter is required.”

  1. Authors’ Responses: The new section 4. The KP in Migraine and Neuropathic Pain provides insight into the KP metabolites in migraine-related mechanisms and neuropathic pain.

  1. Reviewer’s Comment: “Do the authors have a proposal for the use of drugs acting in the KP for chronic pain?”

  1. Authors’ Responses: The use of drugs targeting the KP for chronic pain has been included in a new section, which is titled 5. The KP Compounds as Analgesic Targets in Chronic Pain.

  1. Reviewer’s Comment: “There is no adequate references to related and previous work, e.g. the authors should mention scientific work of the Vécsei L. et al., Körtési T et al.”

  1. Authors’ Responses: The suggested scientific work of Vécsei L. et al. and Körtési T et al. has been included where relevant and referenced throughout the manuscript. We have added additional 9 references by Vécsei et al. and 2 references by Körtési et al. pertinent to the content of our manuscript.

Round 2

Reviewer 2 Report

I would like to thank the authors for addressing my comments.  I recommend accepting the revised  paper.